# Exploring the Wilderness within: An Integrative Metabolomics and Transcriptomics Study on Near-Wild and Colonized *Aedes aegypti*

**DOI:** 10.3390/insects15070507

**Published:** 2024-07-06

**Authors:** Erin Taylor Kelly, Lindsey K. Mack, Geoffrey M. Attardo

**Affiliations:** Department of Entomology and Nematology, College of Agricultural and Environmental Sciences, University of California Davis, One Shields Ave, Davis, CA 95616, USA; etaylorkelly2@gmail.com (E.T.K.); lkmack@ucdavis.edu (L.K.M.)

**Keywords:** *Aedes aegypti*, metabolomics, transcriptomics, pyrethroids, resistance, biomarkers, vector, physiology

## Abstract

**Simple Summary:**

The yellow-fever mosquito, *Aedes aegypti*, is the primary global arboviral vector of dengue, Zika, chikungunya, and yellow fever. Widespread resistance to insecticides has made this mosquito difficult to control. In our study, we compare wild-caught, insecticide-resistant California populations to a susceptible lab colony, Rockefeller, by comprehensively investigating levels of metabolites and conducting comparative gene expression analysis, alongside studies of life history traits. We specifically attempt to identify candidate metabolites that could be investigated further as biomarkers for an insecticide-resistant phenotype. We identify baseline differences in flux through pathways mediating the response to oxidative stress and identify metabolites that vary between the two groups across samples but identify few promising metabolite features with greater than 10-fold change in relative abundance between the strains.

**Abstract:**

This study examines the phenotypic differences between wild-derived F2 Central Valley mosquitoes and the insecticide-susceptible Rockefeller (Rock) lab strain of *Ae. aegypti*. Given the rarity of wild pyrethroid-susceptible populations, the focus of this work is to develop an understanding of the resistance physiology in this invasive mosquito population and explore the potential of metabolites as diagnostic biomarkers for metabolic resistance. This study utilizes metabolomic, gene expression, and lifespan data for a comparison between strains. The findings indicate that wild-derived mosquitoes with greater metabolic resistance have a lifespan sensitivity to restricted larval nutrition. In terms of metabolism and gene expression, Central Valley mosquitoes show increased activity in oxidoreductase, glutathione metabolism, and the pentose phosphate pathway. Conversely, Rock mosquitoes display signs of metabolic inefficiency and mitochondrial dysregulation, likely tolerated due to the consistency and nutritional abundance of a controlled lab environment. The study also examines *Ae. aegypti* P450 and GSTE profiles in relation to other insecticide-resistant groups. While metabolomic data can differentiate our study groups, the challenges in biomarker development arise from few detected markers meeting high fold change thresholds.

## 1. Introduction

Insect reference strains play essential roles in insect research. Reference strains enable reproducible experimentation and can serve as important baselines for comparative analyses. These reference strains differ from colonies in that strains are bred continuously in the lab for many generations without “replenishment” with field-collected mosquitoes [1]. These strains become genetically homogenous and may change significantly as they proliferate without the selective pressures of the field. The Rockefeller (Rock) strain of *Aedes aegypti* (*Ae. aegypti*) has a history of nearly 140 years and is frequently used as a reference strain in insecticide-resistance evaluations of *Ae. aegypti* due to its susceptibility to insecticides typically applied for adult mosquito control, mainly pyrethroids and organophosphates. 

Rock is frequently utilized in insect physiology and resistance studies, where comprehensive physiological research comparing Rock to wild *Ae. aegypti* populations provides important context. Baseline differences in stress response physiology, energy metabolism, and chemoreception have important implications for mosquito research in viral competence, metabolism, and insecticide resistance. In this study, we provide a comprehensive phenotypic comparison of Rock and a near-wild colony derived from the Central Valley of California by integrating metabolomic and transcriptomic analyses with phenotypic assays. Several studies have used transcriptomic data in attempts to identify shared pyrethroid detoxification pathways that could be candidates for surveillance of metabolic resistance [2,3,4,5,6,7]. However, no previous research has integrated metabolomic data. 

California was free of *Ae aegypti* until 2013, when the mosquito was detected in the city of Clovis within Fresno County in the heart of the San Joaquin Valley [8]. Its initial persistence through the Valley’s winter months was a surprise, and it has since been detected throughout the state. There are multiple population groups of *Ae. aegypti* in California. The *Ae. aegypti* in the southern part of the state appear to resemble surrounding populations in the southwestern US, while the origins of *Ae. aegypti* in the San Joaquin Valley are less clear and appear derived from multiple introductions, though one group bears genetic similarities to those found in the Southeastern US [9]. Pesticide deployment to control these populations in Clovis, CA and surrounding cities revealed that they demonstrate a strong resistance to pyrethroids [10,11]. Early eradication efforts failed, and these mosquitoes have remained a persistent problem. This area was even selected as a candidate for the evaluation of a Wolbachia-infected mosquito release program [12].

This study investigates how wild, insecticide-resistant populations of *Ae. aegypti* in California compare to a susceptible lab reference strain (Rockefeller) by integrating transcriptomic and metabolomics analyses. In addition, we use near-wild (F2) populations with similar background genetics and variable resistance profiles to explore hypotheses related to the tradeoffs between resistance and fitness parameters such as lifespan and fecundity. These studies elucidate the potential importance of the pentose phosphate pathway in metabolic resistance and highlight significant alterations in the cellular metabolism between a wild and colonized mosquito line. We also explore the potential for the use of metabolites as markers of the insecticide-resistance phenotype. 

## 2. Materials and Methods

### 2.1. Insect Colonies

Lifespan, fecundity, metabolomic and transcriptomic studies were conducted using near-wild (F2) colonies of *Ae. aegypti* collected from cities in Fresno and Tulare County and maintained in our insectary. The Rockefeller (Rock) mosquitoes are an inbred laboratory strain [1]. The wild-derived colonies are F2 colonies generated from field collections of at least 100 females conducted by the Cornel lab at the Kearney Research and Extension center in three cities in the San Joaquin valley of California; Clovis, Dinuba, and Sanger, in 2018 (Figure 1). This region has a high prevalence of *Ae. aegypti*, and was the site of first detection, in 2013, when *Ae. aegypti* were introduced into the state [13]. 

### 2.2. Mosquito Rearing

Metabolomic and Transcriptomic Analyses: Samples were reared on a standard diet composed of Fluval fish food. Samples were age-matched by pupation date and collected by aspiration 5 days post eclosion. The 10% sucrose solution used to feed adult mosquitoes was withdrawn 36 h prior to sample collection and replaced with water to prevent sugar saturation of analytic equipment. Samples were flash frozen on liquid nitrogen, and then stored at −80 until they were submitted to the West Coast Metabolomics core and the Genomics core for analysis. The collection period for all samples was restricted to a 1.5 h window from 1 to 2:30 p.m. on a single day. 

Lifespan and Fecundity Assays: Study mosquitoes were reared at a density of 200 larvae per tray with 1000 mls of tap water. Larvae were fed two diet treatments consisting of homogenized Fluval Cichlid pellets: our standard *Ae. aegypti* culture diet (full diet) and a restricted diet (half) (Table 1). Four replicates of 200 larvae per diet and strain were maintained in separate cages. Pupation was tracked daily from 5 to 9 days post eclosion. All treatments were blood-fed at 25 days post-eclosion. Adults were placed into cages by tray and dead individuals were counted and removed daily.

### 2.3. Metabolomic Profiling

For metabolomic profiling, 12 pools of 10 adult female mosquitoes were submitted for each of the two lines. Frozen samples were submitted to the University of California, Davis West Coast Metabolomics Center for analysis using a set of 3 complementary metabolomic mass spectrometry (MS)-based assays, designed to measure primary metabolites, lipids, and biogenic amines. Primary metabolites, including carbohydrates, amino acids, fatty acids, nucleotides, and aromatics, were detected using a gas chromatography–time-of-flight (GC-TOF) mass spectrometer (7890 Agilent Gas Chromatograph, Folsom, CA, USA) fitted with a Rtx-5Sil MS (30 m length × 0.25 mm internal diameter with 0.25 μm film made of 95% dimethyl/5%diphenylpolysiloxane) (Restek corporation, Middelburg, ZEELAND, Netherlands). Data acquisition was performed with a Leco Pegasus IV time-of-flight MS instrument (Leco, St. Joseph, MI, USA). Lipidomic analysis was performed by liquid chromatography (LC) (Agilent 6530 Q-TOF LC/MS UPLC) coupled to a quadruple time-of flight (QTOF) charged surface hybrid column (CSH) mass spectrometer (Waters MS Technologies, Manchester, UK). and with MS-Dial 3.98, after filtering for a minimum peak intensity of 1000. Biogenic amines, including acylcarnitines, nucleotides and nucleosides, methylated and acetylated amines, di- and oligopeptides, were measured using a hydrophilic interaction liquid chromatography quadrupole time-of-flight mass spectrometry with tandem mass spectrometry (HILIC QTOF MS/MS) Agilent 6530 Q-TOF LC/MS UPLC, fitted with a Waters Acquity UPLC BEH Amide VanGuard pre-column, Waters Acquity UPLC BEH Amide Column (Waters, Pleasanton, CA, USA). Data was analyzed using Metaboanalyst 5.0 and ChemRich [14,15]. Samples were normalized by the sum of internal standards, log transformed and mean centered prior to performing principal component analysis. For each assay, panel t-tests were performed followed by false-discovery rate adjustment of resulting *p*-values. *p*-Values described in this manuscript refer to FDR-adjusted *p*-values. Accurate peak annotation is a significant hurdle to interpretation of untargeted metabolomics data, so we used the Mummichog algorithm within Metaboanalyst to investigate pathway activity, and generate insight from both annotated and unannotated peaks in our dataset [14]. Mummichog maps peaks to predefined metabolic networks or pathways using retention time and mass-to-charge ratio.

### 2.4. Library Prep and Transcriptome Sequencing 

For transcriptome sequencing, 10 individuals were submitted per our two study populations, RNA was extracted from each individual using a Zymo RNA Cell and Tissue Kit, and submitted to the UC Davis Genome Center for library prep and 3′ Tag-seq analysis. Barcoded sequencing libraries were prepared using the QuantSeq FWD kit (Lexogen, Vienna, Austria) for multiplexed sequencing according to the recommendations of the manufacturer using both the UDI-adapter and UMI Second-Strand Synthesis modules (Lexogen). The fragment size distribution of the libraries was verified via micro-capillary gel electrophoresis on a LabChip GX system (PerkinElmer, Waltham, MA, USA). The libraries were quantified by fluorometry on a Qubit fluorometer (LifeTechnologies, Carlsbad, CA, USA), and pooled in equimolar ratios. The library pool was quantified via qPCR with a Kapa Library Quant kit (Kapa Biosystems/Roche, Basel, Switzerland) on a QuantStudio 5 system (Applied Biosystems, Foster City, CA, USA). The libraries were sequenced on a HiSeq 4000 sequencer (Illumina, San Diego, CA, USA) with single-end 100 bp reads. Reads were checked for quality using FastQC v0.11.9, then trimmed using bbduk, a function within bbmap (v37-50). Resulting reads were aligned to the *Ae. aegypti* LVP_AGWG-50 genome, indexed with an –sjdbOverhang 99 using STAR v2.7.2a. Read files were then indexed using SAMtools v1.3.1. Raw (company names, city, state abbr. if Canada or USA, country) read files are available via the NCBI SRA database under Bioproject #: PRJNA1082311—Whole Body Comparative Transcriptomes of *Ae. aegypti* Strains (Rockefeller Strain versus Clovis California).

### 2.5. Differential Gene Expression and Enrichment Analyses

Differential gene expression analysis was performed using edgeR [16]. Additionally, iDEP (integrated differential expression and pathway analysis) was used for exploratory data analysis [17]. Samples were filtered to only include genes with a minimum of 2 counts per million (CPM) in 12 of 19 libraries. Of the 19,804 genes in 19 samples, 9195 genes passed filtering. Principal component analysis was employed to evaluate sample clustering. The differential gene expression threshold was set at 1.5 minimum fold change, with a false-discovery rate cutoff of 0.05. The differentially expressed genes (DEG) were used for gene-set enrichment analysis (Table 2). PGSEA (parametric gene-set enrichment analysis) was performed using the PGSEA package with all samples [18]. Gene annotations were downloaded from Vectorbase 65. For genes with unspecified products Computed GO Functions and Components were used to infer function, alongside cross referencing of mosquito and drosophilid orthologs. 

## 3. Results

### 3.1. Resistant Lines and Longevity: Nutrient-Stress Effects on Lifespan

The median time-to-knockdown and voltage-gated sodium channel mutation frequency for Central Valley *Ae. aegypti* populations including the strains used in this manuscript are reported in Mack et al. [11], which assayed individuals within one generation of the lines included in this study, and additional toxicological data on Clovis mosquitoes is available in Cornel et al. [10]. The pyrethroid resistance-associated mutations (V410L, F1534C and V1016I) in the voltage-gated sodium channel gene are near fixation in these populations [11,19]. In a susceptible population, the diagnostic time for 100% mortality due to pyrethrum exposure via a CDC bottle bioassay is 15 min. The median knock-down time in response to pyrethrum for Clovis mosquitoes was 82 min, 11 times greater than that of the susceptible reference colony, Rockefeller (5 min). The median knock-down time of Dinuba mosquitoes was 53 min, just 1.35× greater than that of Sanger at 39 min, and 7.5× greater than that of Rockefeller. The median knock-down time of Sanger was 5.5× greater than Rockefeller., The survival of these strains under normal- and reduced-diet regimes was compared to investigate the effect of nutritional deprivation on resistant strains. On a standard larval diet, survivorship was only significantly different by survival analysis for Sanger vs. Rockefeller mosquitoes, with Sanger mosquitoes exhibiting a shorter median survival time but a larger portion of long-lived mosquitoes (>60 days) (Figure 2A), and female body size did not differ significantly between groups (Figure 2B). Females of each population outlived their male counterparts in both treatments. When mosquitoes were subjected to a restricted larval diet, both Sanger and Rockefeller females outlived their standard diet counterparts. This was not the case for the more metabolically resistant Dinuba, which experienced a decrease in life expectancy. For males, the restricted larval diet was only life-extending for Sanger males, and non-significant for Rockefeller and Dinuba. On the full larval diet, we measured first clutch size and observed a slight general trend of decreased fecundity with increasing resistance to pyrethrum, but sample sizes were small and the results were not statistically significant (Appendix B Figure A1).

### 3.2. Metabolomics Panels and Transcriptome Profiles Classify Populations in Principal Component Analysis

Unfortunately, due to the disruptions of the COVID-19 pandemic, we were unable to perform the metabolomic and transcriptomic experiments on the Dinuba and Sanger lines as they were lost during the shutdown. However, the Clovis line was used as it originates in close geographic proximity relative to the origins of the Dinuba and Sanger lines and shares a similar genetic background to other strains collected from the Central Valley [20]. Of the 133 annotated primary metabolite features, 29 were differentially enriched with a fold change of over 1.5, and FDR adjusted *p* < 0.05 (9 up, 20 down in Clovis, relative to Rock). Of the annotated metabolites, only sucrose levels differed by a greater than 10-fold change (up 16× Rock, *p* = 0.001). From the biogenic amine panel, 15 compounds were up and 29 down out of 161 annotated features. The dipeptide Gly-Pro was up 10× in Clovis (*p* < 0.005). From the lipid panel of 590 annotated features, 77 were up, 32 were down in Clovis relative to Rock with a 1.5-fold change cutoff and FDR adjusted *p* < 0.05, and no annotated compounds met a 10-fold change in abundance and sub 0.005 FDR-adjusted *p* threshold. Principal component analysis was performed to investigate sample clustering for transcriptome data and the three metabolomics panels (lipids, biogenic amines and primary metabolites). All four datasets separated the samples by population, with the greatest overlap in the biogenic amine panel (Figure 3A). Gene expression data resulted in the clearest separation by population, but PC1 and PC2 explained just 23.9 and 11.1% of the variance in the data, respectively. Lipid metabolite data, on the other hand, still grouped samples well by population, and PC1 and PC2 explained 43.5% and 18.3%, respectively, of the variance in the dataset (total of 62%) (Figure 3B). Notably, Rock samples appeared to cluster more tightly, likely reflecting the lower diversity in this laboratory strain. We utilized random forest analysis to select features that differentiated between our population groups, the ten top features arranged by feature importance (Figure 3C). From the gene expression data CYP9J26, a cytochrome P450 repeatedly associated with insecticide-resistant groups was a top distinguishing feature [3,5]. Notably, across two assays (primary metabolites and biogenic amines), guanosine and threonine were top features, elevated in Clovis. Phosphatidyl-inositols were distinguishing features abundant in Clovis, while ceramides were enriched in Rockefeller.

### 3.3. Metabolomic Profiles Reveal Enrichment in Pentose Phosphate Pathway Metabolites, Glutathione Metabolism and Lysolipids in Wild Ae. aegypti Relative to Rockefeller

LC-MS peaks were analyzed using the functional analysis module within Metaboanalyst to gain insight from the unannotated metabolites. From the biogenic amine data, 2694 features were analyzed, and 31% were significant with a *p*-value threshold of 0.005. The lipid dataset included 16,841 peaks with 10,556 peaks detected in positive ESI mode, and 6285 detected in negative ESI mode of which 18% and 45% of peaks were significant, respectively. 

We had predicted the maintenance of enzymes conferring pyrethroid resistance like CYPs and GSTs may result in decreased energy stores for Clovis mosquitoes [21], yet we instead observed that Clovis mosquitoes had relative enrichment of saturated and unsaturated triacylglycerols. We observed Clovis mosquitoes had enrichment of unsaturated fatty acids (arachidonic acid being the key compound, Appendix A), but lower amounts of ceramides and phosphatidylethanolamines, which play essential roles in the modulation of membrane fluidity in insect cells and mitochondia [22]. Ceramides, enriched in Rock, also play important roles in mediating fecundity in insects, and are associated with the downregulation of mitochondrial activity and mitophagy in mammals [23,24]. 

Fatty acids and lysolipids, common stress biomarkers, were enriched in Clovis, which aligned with our hypothesis that Clovis may have elevated markers of oxidative stress [25,26], as were levels of oxidized glutathione (Figure 4). Histidine was enriched in Clovis, and plays an important role in normal mosquito egg development [27]. We observed significant under-enrichment of amino acids in Clovis, with threonine, histidine, proline and lysine as exceptions. Differential enrichment of certain B vitamins and their derivatives was also observed, with biotin and folinic acid enriched in Rock, while nicotinamide and 4-pyridoxic acid were enriched in Clovis. We observed subtle alterations in sugar profiles; sucrose and ribose were elevated in Rock (1.6 FC and 1.4 FC), while glucose was very slightly elevated in Clovis (1.3 FC) and glucose-6-phsophate, fructose-6-phosphate, ribose-5-phosphate, phosphogluconic acid and fructose-1-phoshphate, metabolites in glycolysis and the PPP, were all elevated in Clovis (Figure 4).

We observed a differential enrichment of several neuro-active metabolites and urea-cycle metabolites (Figure 4). Histamine levels were moderately elevated in Clovis (1.6 FC, *p* < 0.0005), and histamine acts as a neurotransmitter in insects, with histamine receptors active in mosquito brains and peripheral tissues [28]. We found 3-Hydroxykynurenine was elevated in Rock (2.5 FC, *p* < 0.005), as was kynurenic acid (1.6 FC, *p* < 0.005), both important metabolites of tryptophan metabolism to xanthurenic acid (1.2 FC up in Rock, *p* = 0.006) in mosquitoes, a process essential for normal eye development and mediating oxidative stress from blood-feeding in mosquitoes [29,30]. Gamma-aminobutyric acid (GABA) plays an important role in mediating immunity to dengue infection, and was enriched in Rock (GABA, 1.6 FC, *p* < 0.005) [31]. Components of the urea cycle including ornithine (1.5 FC, *p* < 0.005) and urea (2.2 FC, *p* < 0.005) were moderately elevated in Rock.

Chemical group enrichments are represented in Figure 5. Metabolic networks were relatively less-well annotated with regard to metabolites, and pooling metabolomics panels resulted in fewer significant pathway hits relative to gene-set enrichment analysis. 

### 3.4. Genes Associated with Detoxification Are Overexpressed in Clovis, While Immune and Catabolic Processes Are up in Rock 

Sequencing generated 3′ Tag-Seq single end reads, with an average library size of 1,498,609 reads (min: 826,843, max: 1,928,446) across samples, with an average of 87% of reads mapped to the reference genome. Gene annotations were derived from Vectorbase (Release 65). Over 900 genes (493 up in Clovis, 419 down) were differentially expressed between the two groups with an FDR cutoff of 0.05, and a minimum fold change of 1.5. In addition, 383 (204 up, 179 down) were differentially expressed with a 2-fold-change threshold. Detoxification genes, particularly Cytochrome P450s, were among the most differentially expressed genes (DEGs). The most overexpressed in Clovis was CYP6AG4 (*p* < 0.005, 29 FC), which was associated with a pyrethroid susceptible strain in [5], while CYP9J26 (*p* < 0.005, 17 FC) was second. Others included CYP6AG7 (*p* < 0.005, 7 FC) associated with deltamethrin resistance [7], CYP6BB2 (*p* < 0.005, 5 FC) overexpressed in permethrin, imidacloprid and propoxur selected resistant larvae [2] and insecticide-resistant mosquitoes in Puerto Rico relative to Rock [32]. Additionally CYP6Z8 (5 FC, *p* < 0.005) and GSTE6,4,3 (*p* < 0.005) were overexpressed in Clovis and trended towards enrichment in resistant groups in [3]. GSTE6 was also enriched in Puerto Rican mosquitoes that survived lambda-cyhalothrin exposure [32].

Enrichment analyses (Table 2, Figure 6) reveal that genes related to monooxygenase activity, antioxidant activity and response to oxidative stress are upregulated in Clovis. Additionally, enrichment analyses, complemented by metabolomic data, support enrichment of the pentose phosphate pathway and NADP metabolic processes. The Central Valley *Ae. aegypti* were the subject of a thorough study investigating transcriptional response over time to pyrethroid challenge via bottle bioassay. There is little overlap in the detoxifying genes identified in the study, apart from AAEL006829, a microsomal glutathione-s-transferase. However, there was significant overlap in pathway enrichment (Figure 2 of [33]). This may reflect the fact that in this study the mosquitoes were not challenged with pyrethroids prior to analysis and that the detoxifying genes upregulated in our study relative to the Rock strain are constitutive, providing a protective baseline, playing other roles such as xenobiotic cytotoxic stress mediation, and/or maintenance of the metabolic resistance phenotype. 

We hypothesized that some of the differences we may observe in Clovis and Rock may be related to environmental adaptation. Rockefeller, as a reference strain, is typically maintained in high-temperature, high-humidity insectaries. Our Clovis mosquitoes, on the other hand, are near-wild mosquitoes collected from the Central Valley, USDA zone 9b and parent generations experienced hot, dry summers and winter lows reaching 20–25 degrees Celsius. Recent work investigating the genomic signatures of local adaptation in CA *Ae. aegypti* resulted in a list of 112 candidate genes as putative candidates of local adaptation [34]. Of these, 18 were differentially expressed as transcripts in our study, 11 up in Clovis (*p* < 0.05) and 7 were up in Rock. Up in Clovis were Synotropin-like 1, involved in adapting cellular homeostasis [35] (AAEL019820, 2.2 FC), and fringe, which is involved in modulating Notch signaling [36] (AAEL002253), lipophorin receptor 2 (AAEL019755), unpaired 3, involved in tissue repair and development [37] (AAEL024562), SoxNeuro, a transcription factor involved in central nervous system development, (AAEL000584) were all 1.6-fold up. Bloated tubules (AAEL010883) was up just 1.2-fold, but notably encodes a member of the sodium- and chloride-dependent neurotransmitter family. In Rock, javen-like (AAEL004209), Rab23 (AAEL001532) and Tenascin major (AAEL000405) were up (FC of 1.4, 1.3 and 1.3) and notably all are involved in embryonic development. We identified additional genes involved in ion balance not identified in the local adaptation study, including the differential expression of inward-rectifying potassium-channel genes, with Kir2B up in Clovis (1.8 FC, *p* = 0.001) and Kir2A up in Rock (1.4 FC, 0.01), and AAEL005575, a putative transient receptor potential channel 4, (3.1 FC, *p* < 0.005). 

In Rock, peptidases, cholesterol transport and genes involved in nucleotide and lipid catabolic processes were upregulated. Many of the top upregulated genes in Rockefeller were unspecified products, with computed GO functions as structural components of the cuticle (AAEL020471, 11 FC *p* = 0.005), chitin binding (AAEL023490, 3.7 FC, *p* = 0.007) and multiple predicted serine endopeptidases and protein kinases. Antimicrobial genes cecropin (AAEL029047, 3.7 FC, *p* = 0.02) and defensin antimicrobial peptide (AAEL003832, 3.7 FC, *p* = 0.007) were also upregulated, along with the leucine-rich immune proteins (LRIM) 8, 10A, 10B, 13, 17 and 24, though LRIM18 was up in Clovis. Mitochondrial genes were highly differentially expressed (ND6 11 FC, mRpL37 7 FC). While differentially expressed genes were generally dispersed throughout the genome, using the iDEP’s Genome tool, we found a significant cluster of DEGs on the mitochondria genome. We also observed mild upregulation of genes associated with differentially enriched metabolites such as AAEL012955, a phosphatidylethanolamine binding protein (2 FC, *p* < 0.005), and a sucrose transport protein (AAEL011519, 2 FC, *p* < 0.005). Additionally we found a protein phosphatase-2a (AAEL004288, 1.5 FC, *p* <0.005), perhaps related to the elevated ceramide levels observed in Rock.

### 3.5. Metabolites Clarify Pathway Level Gene Expression Differences in Essential Metabolic Processes and Nervous System Organization

Pathway analysis and metabolite enrichment overlap in hits on the pentose phosphate pathway, with transaldolase and transketolase up in Clovis, and the enrichment of metabolites throughout the pathway (Figure 6B). Clovis mosquitoes may be using the non-oxidative branch to increase flux through glycolysis to the TCA cycle, though these pathways are not as ubiquitously altered as the PPP. Within glycolysis, the phosphopyruvate hydratase complex (AAEL024228, 3.5 FC, *p* < 0.005) and an NAD+ dependent aldehyde dehydrogenase (AAEL01480, 2.8 FC, *p* < 0.005) were up in Clovis, while in the TCA cycle we only saw a mild alteration of malate dehydrogenase which catalyzes the malate to oxaloacetate step (AAEL008166, 1.4 FC, *p* = 0.04), and the isocitrate to oxalosuccinate conversion which precedes amino acid metabolic pathways (AAEL000746, 1.4 FC, *p* = 0.002) though this enzyme also acts in the glutathione metabolism.

In Clovis, we observed hormone changes relative to Rock, particularly farnesol dehydrogenase activity, potentially indicating relatively higher levels of JH synthesis [38], while in Rock, we saw evidence of elevated levels of 20E based on the elevated expression of AAEL027264 (2.4 FC, *p* < 0.005), a putative Phantom (CYP306a1) ortholog. The balance of these hormones can mediate fecundity and metabolic flux [39,40]. In Rock, translation initiation complexes (eIF3h, 2 FC, *p* < 0.005) are active along with lipid transport and localization processes (Figure 6A). AAEL007899, found to be up in non-bloodfed ovaries, was up slightly in Rock (1.4 FC, *p* = 0.04) [28]. Lysosomal activity, mannosidase activity and mitochondrial activity were all enriched in Rock relative to Clovis, potentially representing the breakdown of materials to liberate cellular resources, potentially for reproduction (Figure 6A).

## 4. Discussion

In this study, we combine lifespan data, transcriptomic, and metabolomic assays to provide a thorough phenotypic comparison of Rockefeller and California populations of wild *Ae. aegypti.* We observed differences in levels of metabolic enzymes associated with pyrethroid resistance, and fundamental alterations in metabolic pathways mediating lifespan and response to oxidative stress (Figure 4, Figure 5 and Figure 6). In lifespan assays, when comparing our wild populations with conserved V410L, 1016, and 1534 genotypes, we observed wild pyrethroid-tolerant groups to have modestly longer lifespans, and for females, the lifespan was extended by larval diet restriction, with the exception of our more metabolically resistant group. We did not observe statistically significant differences in fecundity, but the blood-meal timing and small sample sizes may have impacted clutch sizes. These results shed light on how nutrition may modulate the impact of pyrethroid resistance on longevity, as previous reviews have reported variable relationships between pyrethroid resistance and adult longevity [41]. Notably, previous work that isolated the Val1016Ile and Phe1534Cys KDR mutations found little impact on adult longevity [42], while studies incorporating comparisons of KDR mutations and CYP-mediated resistance phenotypes found significant impacts on longevity [41,43]. We speculate that the pathways that mediate the oxidative effects of constitutive maintenance of CYPs and GSTs involved in metabolic pyrethroid resistance can be life-extending when nutritional conditions are favorable.

While restricted diets have life-extending impacts for a wide variety of organisms [44], in drosophila, the amino acid balance is found to modulate this dietary effect, with methionine supplementation alone supporting prolonged lifespan and undiminished fecundity [45]. We found amino acids generally enriched in Rockefeller mosquitoes, particularly methionine, and speculate that the balance of these amino acids may be under unique selective pressure in lab environments, and groups naturally select for high fecundity in lab-mosquito strains.

We hypothesized that we would observe baseline differences in the expression of transcripts of enzymes associated with pyrethroid resistance, such as cytochrome P450s, GSTs and esterases, based on substantial prior research associating these with insecticide resistance [3,32]. Additionally, we predicted these enzymes may raise the oxidative state of the insect, which may be compensated with alterations in the antioxidant pathways to combat oxidative stress. We found evidence for these hypotheses at the metabolite, transcript and phenotype level. We saw an elevated glutathione metabolism and antioxidant activity (Figure 6A, Appendix A) as well as greater activity in the pentose phosphate pathways, an essential source of NADPH required to “recharge” oxidized CYPs and glutathione. Metabolite pathway analysis can be challenging as many metabolites play important roles in multiple pathways, and metabolite level annotations in metabolic pathways are lacking. The transcript data allow us to better explore the sources of differential metabolite levels. Pathways like the PPP, essential to mediating oxidative stress (Figure 4) illustrate agreement between the datasets.

In Rockefeller mosquitoes, the pathways involved in protein turnover and cellular transport and communication are significantly upregulated. Colonization in laboratories removes the pressure from adult mosquitoes to be resilient to significant alterations in environmental conditions such as temperature and humidity. Laboratory colonization may remove the pressure to maintain efficient cellular processes, as calorically rich diets are continuously available, and mates and laying substrates located conveniently nearby. Possible evidence of metabolic dysregulation in our study includes high rates of catabolism and mitochondrial activity (Figure 6A). In humans, ceramides play diverse regulatory roles, stimulating the uptake of free fatty acids, triggering autophagy, and can trigger mitochondrial fragmentation and reduced efficiency [24,46] and may also have an impact on our observed differential mitochondrial gene expression.

Relative to Clovis, Rock appears to have lower levels of JH synthesis at the point of collection, and alterations in insect hormone biosynthesis are indeed identified at both metabolite and transcript levels (Figure 6A,B). In adult insects, JH supports energy storage, perhaps contributing to the TAG enrichment observed in Clovis [47]. Our detection of differential lipid profiles and flux through JH and 20E synthetic pathways may reflect modest alterations in early adulthood, pre-blood meal development. It is interesting to note that both tryptophan metabolism and phosphatidylethanolamine homeostasis play essential roles in insect eye health and development, and metabolites within these pathways are differentially regulated between our two populations [29,48].

We report novel differences in transcripts related to synapse organization and ion balance, which may be compensatory mechanisms of resistance to pyrethroid and other nerve-targeted xenobiotics. We also observed elevated histamine levels in Clovis, and histamine receptors have been found to operate in mosquito brains and peripheral tissues, though histamine receptors were not differentially expressed in our study [28], nor was the voltage-gated sodium-channel transcript in our populations.

We did not identify metabolites that met our conditions of strong differential detection (>10×) consistently across sample groups, but we did identify features with more modest fold changes that reliably classified Rock and Clovis (Figure 3C) including the amino acids serine, threonine and homoarginine. Additionally guanosine, nicotinic acid and histamine represent interesting targets for further investigation (Figure 3C). We also identified enrichment of lipid groups including phosphatidyl inositols, fatty acids and lysolipids whose correlation with resistance may warrant further study.

Taken together, we saw that Rockefeller and our wild Clovis mosquitoes demonstrated robust alterations in fundamental metabolic pathways. While our study cannot conclusively attribute differences to specific aspects of life history, it does represent the first inclusion of metabolomic data in a baseline comparison of mosquito populations, and we sought to pilot an exploration of whether metabolites may present viable biomarkers of phenotypes like metabolic pyrethroid resistance, by identifying features that may be altered broadly across a phenotype despite unique gene-set alterations (such as unique resistance-conferring cytochrome 450 profiles). We found few markers with the high (>10-fold) changes that would best support this aim, but describe interesting metabolic signatures of each population and demonstrate clearly that metabolomic information can powerfully clarify the downstream impacts of differential gene-expression data.

## 5. Conclusions

In this work, we found Central Valley mosquitoes relative to the lab reference strain, Rock, had an elevated expression of enzymes associated with pyrethroid resistance including CYPs, GSTs, etc., and an enrichment of triacylglycerides, fatty acids, lysolipids and nucleotides. In the Central Valley mosquitoes, antioxidant pathways appear to be constitutively upregulated, which may play important roles in mediating context-dependent pyrethroid-related fitness costs. Rock showed evidence of increases in proteolytic pathways and significant alterations in the mitochondrial metabolism relative to our wild population, which may support fertility and/or reflect inefficiencies in the cellular metabolism that may have arisen from laboratory colonization.

## Figures and Tables

**Figure 1 insects-15-00507-f001:**
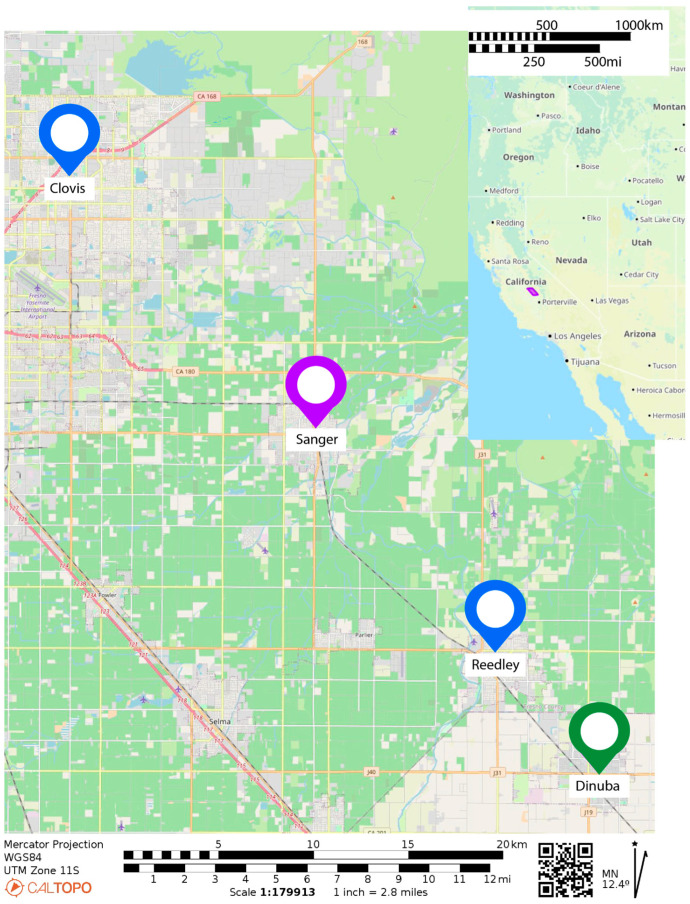
Mosquito regional collection map. Mosquitoes were collected at sites throughout the annotated cities in the summer (July–September) of 2018.

**Figure 2 insects-15-00507-f002:**
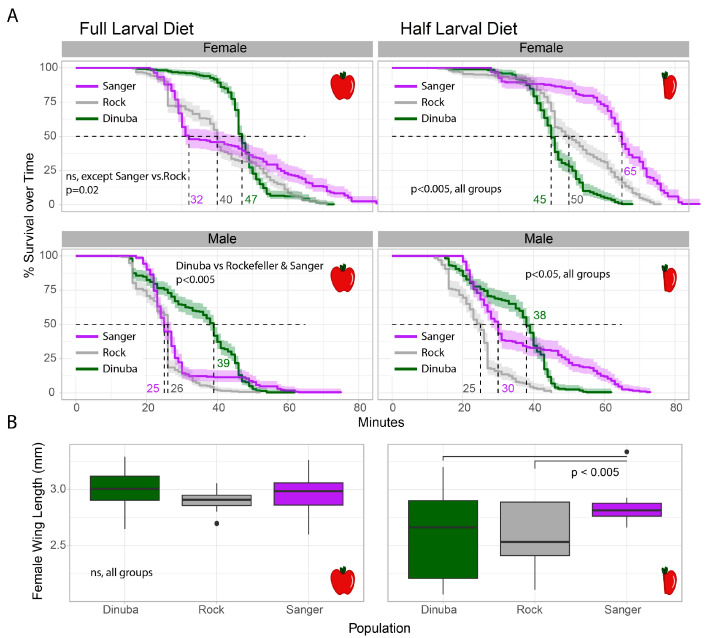
Dietary impacts on lifespan and body size for near-wild *Ae. aegypti* with variable resistance phenotypes. The apple icons represent dietary treatment: a whole apple represents mosquitoes treated with a full larval diet; a half apple represents mosquitoes reared on a half larval diet. (**A**) Lifespan analysis of Rockefeller, Sanger, and Dinuba strains of *Ae. aegypti* under normal and restricted larval dietary regimes. (Rock strain—grey, Dinuba strain—green, Sanger strain—purple). (**B**) Wing lengths of female mosquitoes under normal and restricted dietary conditions. Statistical analysis for life span was carried out using log-rank survival analysis, with Hochberg correction for multiple tests. Wing lengths were tested using a two-way ANOVA followed by Tukey HSD. *p*-Values on graphs represent differences within graph quadrants only. Males had significantly shorter lifespans than female counterparts for all populations, and restricted diet reduced female body size significantly for all groups (*p* < 0.0005). Further results are described in the text.

**Figure 3 insects-15-00507-f003:**
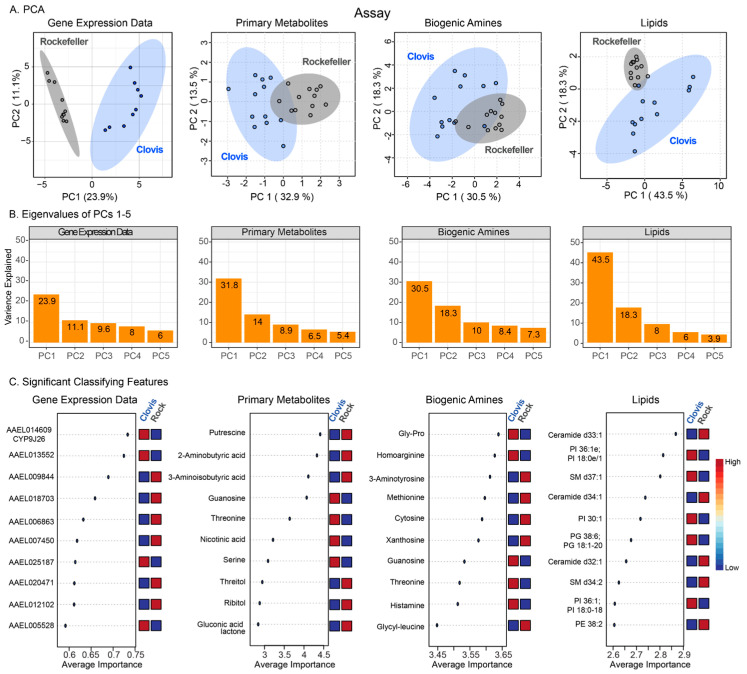
(**A**) Principal component analysis of metabolomic assays and transcriptome data, (**B**) scree plots and (**C**) top 10 classifying features by random forest analysis.

**Figure 4 insects-15-00507-f004:**
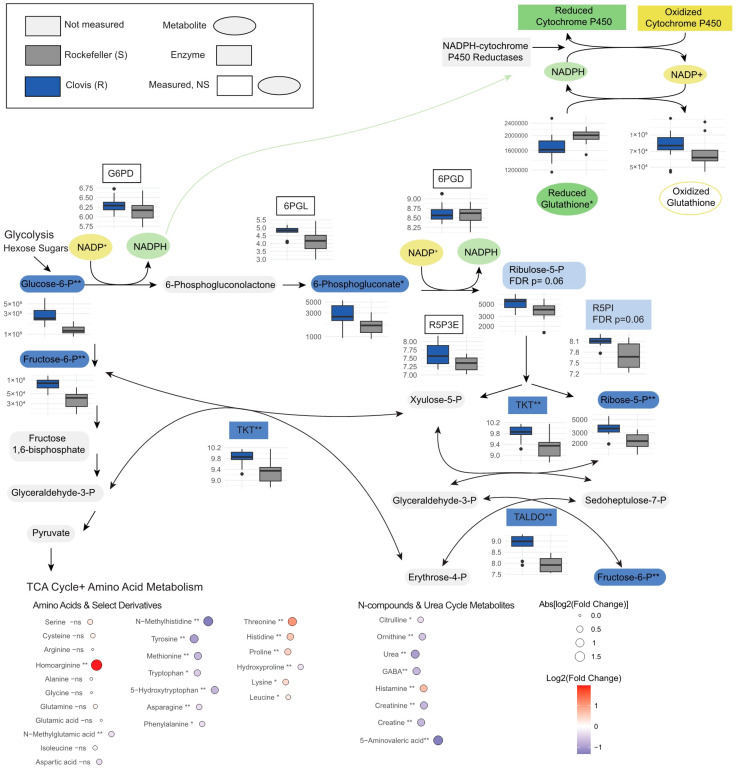
Differential expression of pentose-phosphate pathway genes and metabolites, and downstream metabolite features. * Indicated an FDR-adjusted *p* between 0.05 and 0.005, ** indicates FDR-adjusted *p* below 0.005. ns indicates non-significant difference between treatments. Dots outside of the whiskers in the boxplots represent outlier datapoints.

**Figure 5 insects-15-00507-f005:**
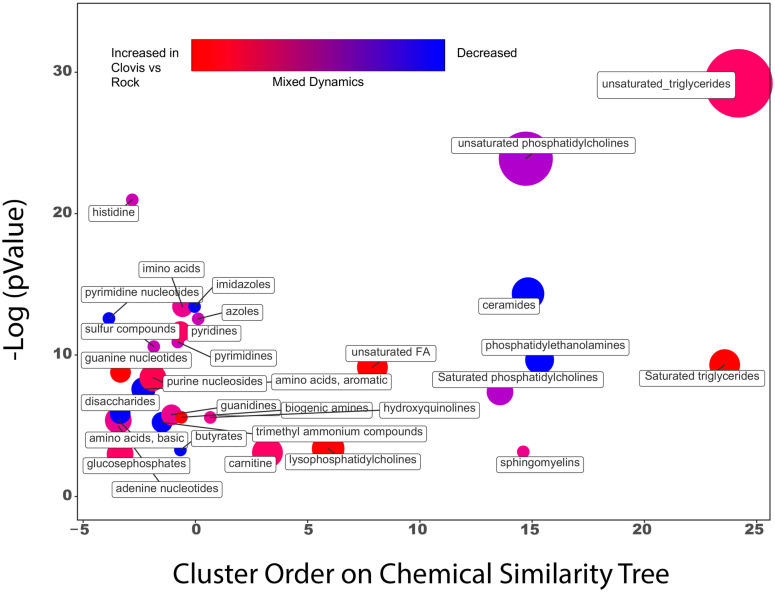
ChemRICH metabolite set enrichment plot for Clovis vs. Rock. Metabolites are classified into chemical classes and evaluated for significance at the set level using the Kolmogorov–Smirnov test.

**Figure 6 insects-15-00507-f006:**
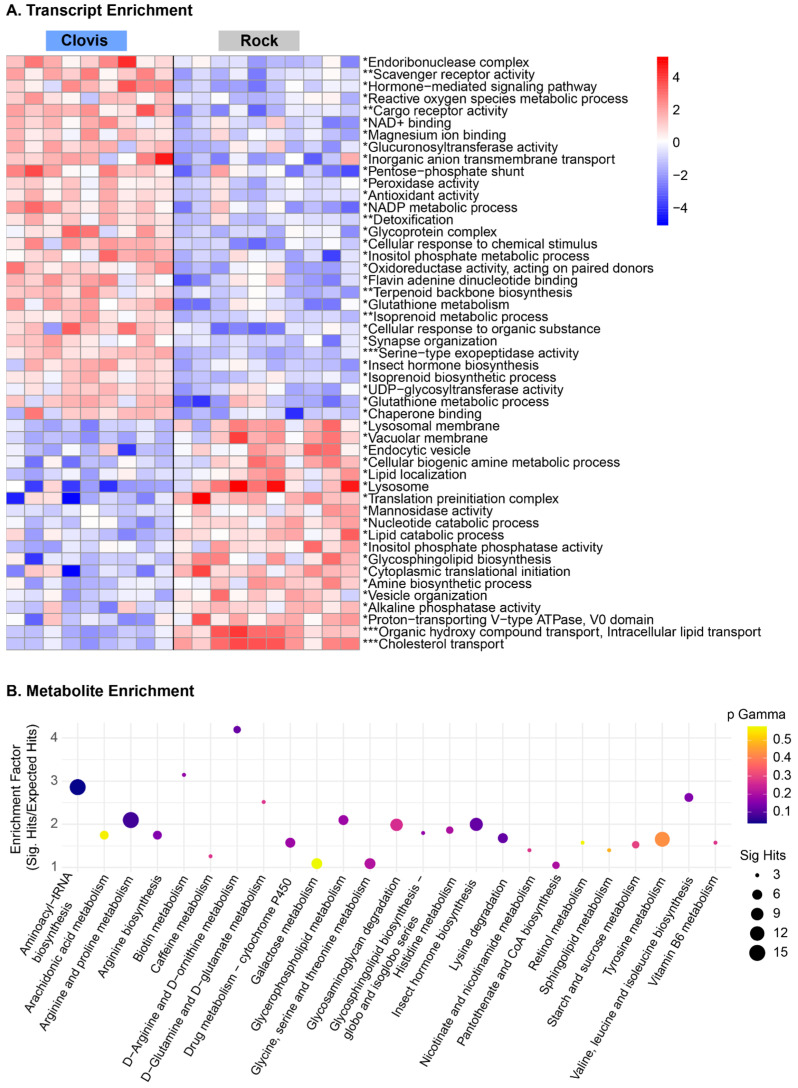
Gene and metabolite enrichment plot. (**A**) Represents enrichment from PGSEA (parametric gene-set enrichment analysis) with an FDR cutoff of 0.05. * represents < 0.05 FDR, ** < 0.005 FDR, *** <0.001. (**B**) Represents metabolite enrichment with an FDR-adjusted cutoff of 0.05.

**Table 1 insects-15-00507-t001:** Larval diet treatments for lifespan and fecundity assays.

Day	Full Diet (mg)	Half Diet (mg)
**1**	372 mg (2 pinches)	186 mg (1 pinch)
**2**	46.5 mg (1 drop)	No food
**3**	93 mg (2 drops)	46.5 mg (1 drop)
**4**	186 mg (4 drops)	93 mg (2 drops)
**5**	186 mg (4 drops)	93 mg (2 drops)
**6**	372 mg (8 drops)	186 mg (4 drops)
**7**	279 mg (6 drops)	139.5 mg (3 drops)
**8**	186 mg (4 drops)	93 mg (2 drops)

**Table 2 insects-15-00507-t002:** Transcriptome pathway enrichment results.

GO Molecular Function			
Direction	Adj. Pvalue	Genes (n)	Pathways
Up in Rock	3.8 × 10^−4^	21	Peptidase activity
	2.6 × 10^−3^	25	Transition metal ion binding
	6.6 × 10^−3^	34	Hydrolase activity
Up in Clovis	6.5 × 10^−5^	13	Tetrapyrrole binding/iron binding/monooxygenase activity
	2.4 × 10^−3^	2	Farnesol dehydrogenase activity
	3.1 × 10^−3^	28	Transition metal ion binding/oxidoreductase activity
**GO Biological Processes**			
Up in Rock	2.3 × 10^−3^	21	Proteolysis
	2.3 × 10^−3^	3	Sterol transport, intracellular lipid transport
	2.6 × 10^−3^	3	Defense response to bacterium

## Data Availability

Raw read files from RNA-seq datasets are available via the NCBI SRA database under Bioproject #: PRJNA1082311—Whole Body Comparative Transcriptomes of *Ae. aegypti* Strains (Rockefeller Strain versus Clovis California): https://www.ncbi.nlm.nih.gov/bioproject/?term=PRJNA1082311 (accessed on 29 February 2024).

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
