# Peer review of "Exploring the Wilderness within: An Integrative Metabolomics and Transcriptomics Study on Near-Wild and Colonized Aedes aegypti"

_insects, 2024, doi:10.3390/insects15070507_

Round 1

Reviewer 1 Report

Comments and Suggestions for Authors

The manuscript by Kelly et al. attempts to compare life history traits, transcriptomics, and metabolomics of wild insecticide resistant populations of Aedes aegypti with a standard susceptible laboratory reference strain (Rock). Previous studies have not integrated metabolomics into comparisons between resistant and susceptible mosquito populations and thus the present study is novel.  The authors find differences in life span associated with larval dietary regimes and in global metabolite and gene expression profiles.  Metabolite analyses revealed quantitatively small differences between resistant and susceptible mosquitoes, but gene expression differences revealed strong differences in certain metabolic pathways between resistant and susceptible mosquitoes, some of which were consistent with the different metabolite profiles identified.  The primary weakness of the study is the lack of consistency between the resistant strains used in the life history vs. metabolomics/transcriptomics, which limits the ability to integrate those sets of data. In addition, the manuscript could use improvements to its presentation which are described in the specific comments below.  Although the authors could not find strong evidence for strong metabolomic signatures of resistance that could be used as reliable biomarkers in future studies, the research provides novel insights into resistance and its mechanisms in emerging populations of an important disease vector in California.

Line 22—Add ‘of’ after ‘understanding’

Line 71—Use of ‘physiologically’ seems misleading as there is very little physiology performed.  Likewise, this is mentioned in the abstract.  The data provide insights into the resistance physiology or generate hypotheses for testing physiological differences, but do not measure it directly.  Thus, the authors may want to change the language to align with experimental data collected, which may help reader comprehension. 

Line 79—In the methods I was not able to find information on sample sizes, numbers of biological replicates, etc.  used for the various longevity assays and the metabolomics/transcriptomics.  This is important information to add.

Line 99—The authors use pellets, but the diet is described in Table 1 as ‘pinches’ or ‘drops’.  Were the pellets pulverized for allowing pinches and were they solubilized to allow for drops?  Please clarify. 

Line 111-  Define ‘TMAO’

Lines 158-170:  These sentences appear to be describing data from a previous study.  This info should be moved to the introduction as it is unclear which data are collected by the present study.  Presumably the relative resistance data are from a previous study? 

Lines 170-174:  Figure 2B is not referenced.  Also, why are body size data mentioned first in the text when they listed second in the Figure.  I would suggest presenting results in the same order between text and figures to avoid reader confusion.

Lines 170-180—Why was the Clovis population excluded from these experiments?

Figure 2A—The female data suggest Dinuba is responding differently from Sanger and Rock, but stats suggest Sanger and Rock are different.  Please confirm this trend. 

Line 216:  Figure 5 is referenced but should this be Figure 4? 

Lines 291-304:  In general the results section contains several examples of discussion, which distracts from the presentation of results.  Comparisons to previous studies and interpretations would be better left for the Discussion.  The lines indicated show an example of where results and discussion are mixed. 

Lines 349-350:  This sentence seems gratuitous and is redundant with info in the introduction.

Line 402:  The potential connection to eye health seems obscure and distracting.  What relevance does eye health have to resistance or any aspect of mosquito biology? 

Comments on the Quality of English Language

Overall is fine. There were a few missing words and mixups with plural/singular usage. 

Author Response

We very much appreciate your time and consideration in evaluating our manuscript. We realize that there is inconsistency in terms of the life history and metabolomics/transcriptomics datasets in terms of the strains used. Unfortunately, due to constraints associated with the pandemic, we were unable to maintain the lines we had performed the life history work on and had to switch to the Clovis mosquitoes, which come from the same broad geographical region of the Central Valley of California. We have updated the text with descriptions of and references to prior published papers in which the toxicological and genetic characteristics of the lines used in this paper are characterized. We hope this will provide context regarding their genetics and resistance phenotypes.

Point by point responses:

Line 22—Add ‘of’ after ‘understanding’

Thank you, “of” has been added at line 22.

Line 71—Use of ‘physiologically’ seems misleading as there is very little physiology performed. Likewise, this is mentioned in the abstract. The data provide insights into the resistance physiology or generate hypotheses for testing physiological differences, but do not measure it directly. Thus, the authors may want to change the language to align with experimental data collected, which may help reader comprehension. 

Thank you, at line 72 we removed physiologically and replaced that language in line 24 of the abstract as well.

Line 79—In the methods I was not able to find information on sample sizes, numbers of biological replicates, etc. used for the various longevity assays and the metabolomics/transcriptomics. This is important information to add.

Thank you for this comment, replicate information has been added to line 103-104, 108 and 131-132.

Line 99—The authors use pellets, but the diet is described in Table 1 as ‘pinches’ or ‘drops’. Were the pellets pulverized for allowing pinches and were they solubilized to allow for drops? Please clarify. 

Thank you, we have added clarifying language to line 102 to reflect that we do homogenize our pellets in a coffee grinder. The measuring spoons used for dispensing the homogenized food come in sizes labeled as “pinches” and “drops” which are equivalent to 186 mg and 46.5 mg respectively. We have revised the dietary table to provide diet amount in milligrams.

Line 111- Define ‘TMAO’

Thank you, we have removed TMAO as the compound is not a metabolite involved in our study, though it can be detected in the metabolite panel.

Lines 158-170: These sentences appear to be describing data from a previous study. This info should be moved to the introduction as it is unclear which data are collected by the present study. Presumably the relative resistance data are from a previous study? 

Thank you for this feedback, we chose to refer to this previous publication as the genetic resistance profile of these lines is characterized in detail in that publication, and the data in that study was collected from the same colonies just one generation off of the mosquitoes used in the lifespan analysis. We have added information to line 167-168 to clarify this.

Lines 170-174: Figure 2B is not referenced. Also, why are body size data mentioned first in the text when they listed second in the Figure. I would suggest presenting results in the same order between text and figures to avoid reader confusion.

Thank you for this comment, the order in which the results are presented was switched to match the figure at line 181, and a reference to Figure 2B is now added to the text at line 182.

Lines 170-180—Why was the Clovis population excluded from these experiments?

Thank you for this comment, Clovis was not included as Clovis and Dinuba had similar levels of pyrethroid resistance from previous research reported in reference 11, and Dinuba was chosen to represent the more resistant phenotype. We had planned to repeat the lifespan and fecundity experiments to include our other lines, but were not able to due to pandemic disruptions. We chose to include the data due to the fact that we were aware that these populations (Clovis, Dinuba, Sanger) had a shared Voltage Gated Sodium Channel mutation set, allowing us to control for that aspect of their biology. Additionally, we were able to compare Rock to these wild lines.

Figure 2A—The female data suggest Dinuba is responding differently from Sanger and Rock, but stats suggest Sanger and Rock are different. Please confirm this trend. 

Thank you for this comment, that is correct- the difference between Sanger and Rock met our significance threshold, while the other comparisons (Dinuba vs. Rock, Dinuba vs. Sanger) did not.

Line 216: Figure 5 is referenced but should this be Figure 4? 

Thank you, we have altered the text at line 228 to state that chemical group enrichments are represented in figure 5, rather than pathway enrichments.

Lines 291-304: In general the results section contains several examples of discussion, which distracts from the presentation of results. Comparisons to previous studies and interpretations would be better left for the Discussion. The lines indicated show an example of where results and discussion are mixed. 

Thank you for this suggestion. We have rewritten this section to remove discussion-like language at line 294 but retained lines 295-302 as we felt they contextualized a hypothesis we chose to investigate with this data: that we may see differential expression in genes previously identified as being associated with environmental adaptation.

Lines 349-350: This sentence seems gratuitous and is redundant with info in the introduction.

Thank you, we have deleted this line and revised it at line 363.

Line 402: The potential connection to eye health seems obscure and distracting. What relevance does eye health have to resistance or any aspect of mosquito biology? 

Thank you for this comment, I have revised the text to remove the reference at line 419.

Reviewer 2 Report

Comments and Suggestions for Authors

The manuscript characterized a lab strain Rock and F2 of field-collected strains in California. The study included lifespan comparisons of three strains and metabolomics and transcriptomics of two strains. The study concluded that wild mosquitoes have greater metabolic resistance to insecticide while they are more sensitive to restricted larval nutrition. The metabolic differences were identified for the increased oxidoreductase, glutathione metabolism, and pentose phosphate pathways in the mosquitoes caught in the field.  Unfortunately, I find the manuscript very difficult to follow in the logical flow of the study aims and in the evidence supporting the conclusion.

The first part of the critique would be the things that may not be fixable because the experiments have already been done.

The pieces of information from each strain, while providing valuable insights, do not compose the overall picture of the study results.  The dietary impact study and knock-down study were done with Rock, Dinuba, and Sanger strains, while metabolomics and transcriptomics were done with Rock vs. Clovis collection without toxicological data. Can we consider all field-collected strains the same? It is very unlikely, based on the differences in the diet impacts on Dinuba and Sanger strains. Therefore, extrapolation of the diet impact results to Clovis strain, which was used for metabolomics and transcriptomics, could not be justified.  Specifically, the lack of toxicological data for the Clovis strain is a critical component lacking in the study. Addressing these issues could significantly enhance the impact of your study. 

In addition, this approach, metabolomics and transcriptomics, by comparing one field collected to a lab mosquito strain will never reach biomarker development considering many complex variable factors in the experiment, including different genetic backgrounds. The major difficulty in accepting authors’ conclusion maybe due to unreasonable levels of extrapolation from a bergy bit.

Other minor comments are;

Aedes aegypti and Ae. are mixed in the overall manuscript.  Same for Rocketeller and Rock.  Please carefully follow the basic rules.

The founder population sizes, and the collection dates need to be added.

The units for larval diet could be metric measures rather than “drops” and “pinches”.

The results (lines 162 and after) showed toxicological data. This could be important information as the study's baseline could be shown as a separate figure with the error bars for the biological replications.

Figure 5 caption needs further details explaining the figure.  What is the X-axis “Median ClogP of Clusters”?  I guess it is a log fold differences calculated by the median of the clusters.  Does it mean higher expression in the Clovis strain?  It does not tell what is compared to what?

Author Response

We very much appreciate your time and consideration in evaluating our manuscript. We realize that the strains used in the life history and metabolomics/transcriptomics datasets are inconsistent. Unfortunately, due to constraints associated with the pandemic, we were unable to maintain the lines on which we had performed the life history work and had to switch to the Clovis mosquitoes, which come from the same broad geographical region of the Central Valley of California. 

We have added language to line 167-168 to reflect that toxicological information and voltage-gated sodium channel data for all populations used in this study is published in a previous manuscript from reference 11, and additional data is available for Clovis in reference 10. Regarding the issues comparing Clovis and Rockefeller to look for potential biomarkers, we agree with the points outlined by the reviewer. We chose to pursue this strategy with the following rationale:

Highly pyrethroid susceptible field lines of Aedes aegypti are rare, and it was necessary to initiate our investigation comparing a truly susceptible group and a population with resistance to pyrethroids that was significant such that meaningful field control with pyrethroids was not possible, which was demonstrated for Clovis for permethrin, sumithrin and pyrethrum in reference 10. Putative biomarkers would have to have a reliably strong difference between any susceptible and resistant group, regardless of other aspects of biology. It was noteworthy then that we did not, with this panel, identify biomarkers with large differences in terms of relative abundance (>10 fold) that could be investigated more robustly. However, we did observe multiple differences in compounds at a smaller scale linked to well-characterized biochemical pathways (such as the pentose phosphate pathway). These findings were reciprocated at the gene expression level with differences in transcript abundance of enzymes functioning in this pathway, which provides more support for these findings and new investigatory pathways to understand the role of these systems in maintaining the phenotype of these mosquitoes.

Other minor comments are;

Aedes aegypti and Ae. are mixed in the overall manuscript. Same for Rocketeller and Rock. Please carefully follow the basic rules.

Thank you for this comment, we have edited the text to use Ae. aegypti throughout the text.

The founder population sizes, and the collection dates need to be added.

Thank you, founder population sizes have been added to line 88.

The units for larval diet could be metric measures rather than “drops” and “pinches”.

Thank you for this comment, and apologies for the confusion. We chose to refer to the measurement in this fashion to reflect the measuring tools used to measure out our food. We have converted these measurement units to milligrams for easier interpretation and replication and updated the associated table in the materials and methods.

The results (lines 162 and after) showed toxicological data. This could be important information as the study's baseline could be shown as a separate figure with the error bars for the biological replications.

Thank you for this comment, we have added a clarifying sentence at line 171-173 that the toxicological data, published in a separate paper, was indeed collected for the populations utilized in this study, within one generation of the individuals used for the lifespan, metabolomic and transcriptomic studies. We have also added a line to Additional toxicological data and field trial data is available for Clovis in reference 10.

Figure 5 caption needs further details explaining the figure. What is the X-axis “Median ClogP of Clusters”? I guess it is a log fold differences calculated by the median of the clusters. Does it mean higher expression in the Clovis strain? It does not tell what is compared to what?

Thank you for this comment, the figure has been revised so that the X axis now says “Cluster order on chemical similarity tree” and caption better describes the X axis.

Updated Caption:

Figure 5. ChemRICH Metabolite Set Enrichment Plot for Clovis Vs Rock. Metabolites are classified into chemical classes and evaluated for significance at the set level using the Kolmogorov-Smirnov test.

Reviewer 3 Report

Comments and Suggestions for Authors

In this study, researchers compared wild-caught, insecticide-resistant California populations of the yellow-fever mosquito Aedes aegypti to a susceptible lab colony, Rockefeller population. They conducted metabolites and transcriptome profiles to identify potential biomarkers for insecticide resistance. The study findings showed baseline differences in pathways related to oxidative stress and identified some varying metabolites between the two groups. The study is intriguing, and currently only a sparse number of prior investigations in this area. However, I have some comments regarding the study design and presentation which I outlined below.

1. Line No: 92-93: The mosquitoes may be experiencing starvation stress due to a lack of sucrose for 36 hours, which may impact transcriptome and metabolome profiles. I am curious about the author’s rationale for subjecting the mosquitoes to a 36 hour starvation period before sample collection. Are the altered metabolites identified in this study indeed involved in insecticide resistance?

2. Line 100-101: I feel blood-feeding at 25 days post-eclosion might be too long. Typically, adult Aedes aegypti mosquitoes start feeding within days of emerging, often within the initial week. Blood feeding in Aedes aegypti can be impacted by temperature; higher temperatures usually prompt earlier feeding, whereas lower temperatures may delay it. It's unclear what temperature conditions these mosquitoes were maintained. If these mosquitoes fed late, it could potentially affect their fecundity rate.

3. Line 186: Is it a FDR-corrected p-value or simply a p-value?

4. Line 196: Please provide the full form of PC1.

5. Table 2: Pathway enrichment results of transcriptome? Please include this information.

6. Is there a correlation between the metabolite data and the transcriptome data? What percentage of altered metabolites correlates with the transcriptome data in each strain? This comparison aids in determining whether transcriptome data alone is sufficient to identify insecticide resistance in the mosquito population, or if both transcriptome and metabolome data need to be analyzed to identify insecticide resistance in mosquito populations. Please discuss this in the discussion section.

7. Based on your data, which altered metabolites serve as markers for identifying insecticide resistance in the Ae. aegypti mosquito population?

Author Response

We appreciate your time and consideration in evaluating our manuscript. Please find point-by-point responses to your queries below.

  1. Line No: 92-93: The mosquitoes may be experiencing starvation stress due to a lack of sucrose for 36 hours, which may impact transcriptome and metabolome profiles. I am curious about the author’s rationale for subjecting the mosquitoes to a 36 hour starvation period before sample collection. Are the altered metabolites identified in this study indeed involved in insecticide resistance?

Thank you for this comment, a clarification for this methodological choice was added to line 100. The metabolomic analyses performed are sensitive to large amounts of simple carbohydrates which can cause problems with the LC and GC /MS systems used for the metabolomics analyses. At the suggestion of the metabolomics facility we incorporated withdrawal of sucrose at 36 hours prior to sample collection. Given that all sampled mosquitoes had the same treatment, their response to the sampling conditions would be influenced by underlying aspects of biology unique to each strain, and resistance status is an aspect of that biology.

  1. Line 100-101: I feel blood-feeding at 25 days post-eclosion might be too long. Typically, adult Aedes aegyptimosquitoes start feeding within days of emerging, often within the initial week. Blood feeding in Aedes aegyptican be impacted by temperature; higher temperatures usually prompt earlier feeding, whereas lower temperatures may delay it. It's unclear what temperature conditions these mosquitoes were maintained. If these mosquitoes fed late, it could potentially affect their fecundity rate.

Thank you for this comment, we appreciate this point and have added to our discussion at lines 383-384 so that readers may also consider that the late feeding may have impacted our results.

  1. Line 186: Is it a FDR-corrected p-value or simply a p-value?

Thank you for this comment; the text has been updated to clarify these are FDR-corrected p-values.

  1. Line 196: Please provide the full form of PC1.

Thank you, this information has been added to line 216.

  1. Table 2: Pathway enrichment results of transcriptome? Please include this information.

Thank you, this has been clarified in the title table in line 345.

  1. Is there a correlation between the metabolite data and the transcriptome data? What percentage of altered metabolites correlates with the transcriptome data in each strain? This comparison aids in determining whether transcriptome data alone is sufficient to identify insecticide resistance in the mosquito population, or if both transcriptome and metabolome data need to be analyzed to identify insecticide resistance in mosquito populations. Please discuss this in the discussion section.

Thank you for this comment. Yes, we initially discussed this in the results at lines 348-369, but selecting a percentage of metabolites that correlate is difficult, as a majority of metabolites do not have well-known pathway annotations, and well-studied metabolites have roles in multiple pathways. We have modified the discussion at lines 411-416 and 428-429 to highlight correlations between the two datasets in terms of functional overlap.

  1. Based on your data, which altered metabolites serve as markers for identifying insecticide resistance in the Ae. aegyptimosquito population?

Thank you for this comment, this topic is discussion in lines 204-226, and we have edited the discussion to further discuss metabolites of interest at lines 445-451

Round 2

Reviewer 1 Report

Comments and Suggestions for Authors

Thank you for addressing my comments.  My only remaining minor suggestion would be to provide a brief explanation that disruptions due to Covid pandemic prevented you from conducting experiments on certain lines.  This would help provide context to the reader as to why experiments were not conducted on some of the lines.  Just a simple statement would suffice.  E.g., 'Due to disruptions of the Covid19 pandemic, we were unable to perform the ??? experiments on the ??? line(s).'

Author Response

Thank you for taking the time to provide constructive comments and for rereviewing our manuscript and for your thoughtful feedback. An explanation of the situation regarding the loss of lines during the pandemic has been added to the paper.

Reviewer 2 Report

Comments and Suggestions for Authors

Significant improvements are made in the revision.  

Author Response

Thank you for your constructive comments and for taking the time to rereview our manuscript. We are happy that you found the paper to be improved.